# The Relationship between Imaging-Based Body Composition Analysis and the Systemic Inflammatory Response in Patients with Cancer: A Systematic Review

**DOI:** 10.3390/cancers11091304

**Published:** 2019-09-04

**Authors:** Tanvir Abbass, Ross D. Dolan, Barry J. Laird, Donald C. McMillan

**Affiliations:** 1Academic Unit of Surgery, School of Medicine, University of Glasgow, Glasgow Royal Infirmary, Glasgow G4 0SF, UK; 2Institute of Genetics and Molecular Medicine, University of Edinburgh, Edinburgh EH4 2XR, UK

**Keywords:** body composition, computed tomography, dual energy X-ray absorptiometry, ultrasound, magnetic resonance imaging, systemic inflammation, cancer, cachexia

## Abstract

*Background and aim:* Cancer is the second leading cause of death globally. Nutritional status (cachexia) and systemic inflammation play a significant role in predicting cancer outcome. The aim of the present review was to examine the relationship between imaging-based body composition and systemic inflammation in patients with cancer. *Methods:* MEDLINE, EMBASE, Cochrane Library and Google Scholar were searched up to 31 March 2019 for published articles using MESH terms cancer, body composition, systemic inflammation, Dual energy X-ray absorptiometry (DEXA), magnetic resonance imaging (MRI), ultrasound sonography (USS) and computed tomography (CT). Studies performed in adult patients with cancer describing the relationship between imaging-based body composition and measures of the systemic inflammatory response were included in this review. *Results:* The literature search retrieved 807 studies and 23 met the final eligibility criteria and consisted of prospective and retrospective cohort studies comprising 11,474 patients. CT was the most common imaging modality used (20 studies) and primary operable (16 studies) and colorectal cancer (10 studies) were the most commonly studied cancers. Low skeletal muscle index (SMI) and systemic inflammation were consistently associated; both had a prognostic value and this relationship between low SMI and systemic inflammation was confirmed in four longitudinal studies. There was also evidence that skeletal muscle density (SMD) and systemic inflammation were associated (9 studies). *Discussion:* The majority of studies examining the relationship between CT based body composition and systemic inflammation were in primary operable diseases and in patients with colorectal cancer. These studies showed that there was a consistent association between low skeletal muscle mass and the presence of a systemic inflammatory response. These findings have important implications for the definition of cancer cachexia and its treatment.

## 1. Introduction

Cancer is the second leading cause of death and has resulted in 9.6 million deaths worldwide in 2018 [1]. Patients present with various stages of cancers and the treatment aim is usually classified as curative or palliative, depending on the stage of the disease and patient factors (performance status, co-morbidities). The decision-making process for each patient is complex and involves multidisciplinary team discussions; moreover, using the optimal therapy in the correct patients improves quality of life and survival and has positive implications for health care resources.

As cancer progresses, it is frequently associated with anorexia, weight loss and loss of skeletal muscle mass (termed cancer cachexia) and these are known to be associated with poor outcome. The basis for such changes in body habitus is not clearly understood [2]. For example, some tumour types, such as lung and gastrointestinal cancers, are particularly associated with weight and muscle loss; however, in other tumour types (e.g., breast, prostate), this is less common. 

While in the past, weight loss and body mass index (BMI) have been used as indicators for malnutrition and cancer cachexia, there have been ongoing attempts to better define body composition in patients with cancer. Various techniques, such as bioelectric impedance analysis, whole body potassium, and air displacement plethysmography, have been used to quantify body composition in the research setting. More recently, imaging-based approaches, such as Dual-energy X-ray absorptiometry (DEXA), magnetic resonance imaging (MRI), ultrasound scan (USS) and computed tomography (CT), have been utilized. These imaging-based body composition measuring modalities have the advantage that they are readily available and would be readily adopted into clinical practice if shown to be clinically useful. In particular, an excellent agreement between DEXA, CT, and MRI for adipose tissue and skeletal muscle has been reported [3,4,5,6,7].

In particular due to its routine use in cancer staging, CT has become the preferred standard for measuring body composition, providing useful new information on body compositional changes associated with cancer cachexia [3,4,5]. In particular, fat and muscle area at Lumbar 3 (L3) vertebra level is highly correlated to other measures of body composition [4,8]. A Skeletal muscle index (SMI) calculated from image based body composition analysis, provides a reliable objective assessment of skeletal muscle quantity [5]. These imaging-based modalities (DEXA,CT,MRI) have also been investigated in various benign diseases, such as myopathies, malnutrition, chronic respiratory, renal and cardiac illnesses, and these have been found to be reliable tools for the assessment of muscle quantity [9].

The basis of the disproportionate loss of skeletal muscle over adipose tissue is not clear. However, it now recognised that systemic inflammatory response is associated with weight and muscle loss and poorer outcomes in patients with cancer [10] and may be useful in identifying the various stages of cachexia [11] (Table 1). Therefore, the routine clinical use of radiological imaging offers the opportunity to examine these relationships in more detail. The present review examines the relationship between imaging-based body composition and systemic inflammatory response in patients with cancer.

## 2. Patients and Methods

### Data Sources and Search Strategy

A study protocol was developed in accordance to the Preferred Reporting Items for Systematic Review and Meta-analyses (PRISMA) guidelines [12]. A systematic search using Medline, EMBASE, Cochrane databases and Google Scholar was carried out to identify studies assessing the relationship between body composition, systemic inflammation and cancer using MESH Terms “body composition, computed tomography (CT), Dual Energy X-ray Absorptiometry (DEXA), Magnetic resonance imaging (MRI), Ultrasound scan (USS), systemic inflammation, cancer and cachexia”. The search was conducted from the start of the relevant database to the date of the last search, which was 31 March 2019.

All relevant studies evaluating the relationship between body composition and systemic inflammatory response in adult patients with cancer were included. For this systematic review, animal studies, conference abstracts, reviews, non-English studies and those not measuring the topic of interest were excluded. The study titles were screened for relevance before a review of abstracts and full texts (TA). Discrepancies were addressed by re-examination and discussion with the senior author (DCM). Reference lists from relevant studies were hand-searched for any other eligible studies. The eligible studies were then assessed for quality using the 22-point STROBE (STrengthening the Reporting of OBservational studies in Epidemiology) checklist, which is a validated methodological quality assessment tool used for submitting studies and to provide feedback by reviewers [13].

## 3. Results

Initially, 807 studies were identified, and following subsequent screening of titles, abstracts and then full papers, 23 met the final eligibility criteria (Figure 1). Articles were excluded if there was no relationship studied between body composition and systemic inflammation (*n* = 192), animal studies (*n* = 141), duplicates (*n* = 17), non-cancerous (*n* = 16), full articles not available (*n* = 3) and those that were reviews only (*n* = 2). Another 411 studies were excluded following review, as they did not address the topic of interest, namely the relationship between imaging-based body composition and systemic inflammation in patients with cancer.

No study examining the relationship between MRI and USS-derived body composition analysis and markers of the systemic inflammatory response was identified. There were three studies that examined the relationship between DEXA-derived body composition analysis and markers of the systemic inflammatory response and 20 studies that examined this relationship with CT-derived body composition analysis. Of the 20 CT studies, 19 reported body composition analysis using the L3 level of the vertebral column.

All DEXA studies [14,15,16] included in this review used LUNAR DPX-L & LUNAR PRODIGY software (Discovery^®^, Hologic, Bedford, MA USA) for body composition measurements. Of the 20 CT studies, six studies [17,18,19,20,21,22] used Slice-O-Matic software (TomoVision, Montreal, Quebec, Canada), three studies [23,24,25] used Image J software (NIH Image J version 1.47, https://imagej.nih.gov/ij/), two studies [26,27] used Synapse Vincent software (Fujifilm Medical, Tokyo, Japan), two studies [28,29] used Infinitt PACS software (INFINITT Healthcare Co., Ltd, Seoul, Korea)**,** one study [30] used OSIRIX software (OSIRIX ^®,^ Version 3.3, downloaded from http://www.osirix-viewer.com), one study [31] used Terrarecon software (Terarecon 3.4.2.11, San Mateo, CA, USA), one study [32] used Somatom Software (Somatom Sensation, Siemens, Fairfield, CT, USA) and manual CT images analyses was performed in four studies [33,34,35,36]. All the 20 CT studies used same thresholds for muscle (−29 to 150 HU) to measure SMA, which were normalized for height ^2^ to define SMI. Irving et al. compared Slice-O-Matic with Image J in 26 obese subjects with intra- and inter-investigator co-efficient with a reliability of R^2^ = 0.99 and a mean difference of less than 2% [37], Richards et al. compared Slice-O-Matic and Image J in a sample of 50 cases with a mean difference of 7.50 cm^2^ [23], Van Vugt et al. compared four software packages (Image J, sliceOmatic, OsiriX and FatSeg) in a sample of 50 cases with inter-software an intra-class correlation coefficient of (≥0.999) and a *p*-value of <0.001 [38], and Teigen et al. compared Slice-O-Matic with Image J in 51 cases with an overall mean difference of 1.53 cm^2^ [39]. Therefore, it appears that there was excellent agreement between the most commonly used software packages. As a result, the study cohorts were considered together in the present review.

Using the STROBE checklist, the breakdown of quality of these studies is given in Table 2. The lowest score achieved was 16 [32] and the highest was 20 (multiple). Length of follow up was a variable. The characteristics of the included studies, the relationship between imaging-based body composition and systemic inflammation are summarized in Table 2. The measurement of body composition was carried out in three studies using DEXA and in 20 studies using CT. Therefore, 23 studies met the final inclusion criteria, with 11,474 cancer patients studied (6281 males and 5193 females).

The majority of the studies were single centre (20 studies, *n* = 8785), prospective (12 studies, *n* = 8611) and carried out in European countries (12 studies, *n* = 3272). There were seven studies carried out in Asian countries (*n* = 2362) and four studies in the USA (*n* = 5840). The majority of studies were in primary operable cancer (16 studies, *n* = 10,198) and colorectal cancer was the most commonly studied cancer (10 studies, *n* = 8344).

The skeletal muscle index (SMI) was most commonly measured (21 studies, *n* = 11,277) and C-reactive protein (CRP) and albumin were the most commonly measured markers of the systemic inflammatory response (18 studies, *n* = 8903 and 23 studies, *n* = 11,474 respectively). A significant inverse relationship between SMI and CRP was reported in 13 studies (*n* = 5201), a significant inverse relationship between SMI and mGPS (combination of CRP and albumin) was reported in eight studies (*n* = 1934), a significant inverse relationship between SMI and NLR was reported in eight studies (*n* = 5717) and a direct relationship between SMI and albumin in 15 studies (*n* = 7002).

A low SMI was reported to be associated with shorter overall survival (10 studies, *n* = 5202) and associated with shorter overall survival independent of markers of the systemic inflammatory response (seven studies, *n* = 4481). When both sarcopenia and systemic inflammation were combined, the risk of death was doubled [20].

Low skeletal muscle density (SMD) and its relationship to systemic inflammation was reported in nine studies (*n* = 6025). A significant inverse relationship between SMD and NLR was reported in seven studies (*n* = 5531), a significant inverse relationship between SMD and mGPS in four studies (*n* = 1509) and a direct relationship between SMD and albumin in six studies (*n* = 1906). A low SMD was reported to be associated with decreased overall survival in four studies (*n* = 1412), cancer-specific survival in two studies (*n* = 533) and disease-free survival in one study (*n* = 211). 

A total of 19 of 23 studies were cross-sectional cohort studies. Four studies were longitudinal cohort (1 in DEXA [15] and three in the CT group [19,20,36]). A significant inverse relationship between SMI and CRP was reported in two longitudinal studies (*n* = 2941), and an inverse relationship between SMI and NLR in two longitudinal studies (*n* = 857) and a direct relationship between SMI and albumin in three longitudinal studies (*n* = 3704).

## 4. Discussion

The results of the present systematic review show that in approximately 10,000 patients with cancer, there was a consistent association between CT-derived SMI/SMD and systemic inflammation, as evidenced by CRP, albumin (mGPS) and Neutrophil Lymphocyte Ratio (NLR). To our knowledge, this is the first such systematic review. Since this relationship was determined mainly in cross-sectional studies and in primary operable cancers, it is not clear whether a low SMI/SMD results in the presence of systemic inflammation or whether the presence of systemic inflammation results in low SMI/SMD. Nevertheless, given the importance of these respective measures in defining the syndrome of cancer cachexia and cancer progression, it is important to examine this relationship in more detail, particularly in patients with advanced cancer [10,11,40,41]. CT abdomen is part of cancer staging in patients with a wide variety of cancers, including gastrointestinal, hepato-biliary, pancreatic, renal, bladder and lung cancers. From CT abdomen, the L3 level can be readily calculated using manual or semi-automated software packages and using muscle and adipose tissue thresholds, all components of body composition can be calculated. 

However, the clinical utility of landmarks other than L3 is not clear. There is some debate as to whether the measurement of psoas muscle at lumbar 3 level is less reliable and inferior to measuring all muscles at this level [42,43] and therefore, these studies [44,45] were considered separately. Using psoas muscle measurement, Hervochon and co-workers, in a cohort of 161 patients with operable NSCLC, reported that low SMI (total psoas area ≤ 33rd percentile) was significantly associated with elevated CRP [44]. Furthermore, Okugawa and co-workers, in a cohort of 308 patients with operable CRC, reported that low SMI (using sex-specific median values of psoas muscle index, male: 286.8 mm^2^/m^2^, female: 210.6 mm^2^/m^2^) was significantly associated with elevated CRP and low albumin [45]. Therefore, it would appear that skeletal muscle, however, assessed from CT scans, is consistently associated with measures of the systemic inflammatory response.

Since there is little evidence that increasing skeletal muscle mass is associated with a reduction in cancer-associated systemic inflammation, a plausible hypothesis explaining this relationship is that a pro-inflammatory state is the main etiological factor in progressive muscle loss and this underpins the nutritional and functional decline associated with cancer cachexia. For example, comparing inoperable cancer with operable cancer, the former is consistently associated with greater tumour bulk and greater elevation of the mGPS and NLR [40,41] and weight and skeletal muscle loss is a feature of the cachexia of advanced disease. Furthermore, a greater elevation of the mGPS is associated with more aggressive tumours, such as lung and pancreatic cancer [40,46,47], and these tumours are characterized as the tumour types most commonly associated with cachexia.

Therefore, it is of interest that there is good evidence that elevated circulating concentrations of key pro-inflammatory cytokines (e.g., Interleukin 6 [IL-6], Interleukin 1 [IL-1]) link the presence and aggressiveness of the tumour to the loss of skeletal muscle mass [48,49] and elevated markers of the systemic inflammatory response [50]. If this was the case, the pro-inflammatory state could be expected to be a catabolic event and would predate the significant loss of skeletal muscle mass. Indeed, of the longitudinal studies reviewed, the presence of a systemic inflammatory response at baseline was associated with lower SMI on follow-up independent of tumour stage in patients with primary operable cancer [19,20]. Furthermore, it is recognized that an elevated CRP and low albumin concentration are risk factors for the development of cancer [51,52]. Taken together, these observations directly link the loss of skeletal muscle mass and the presence of a systemic inflammatory response. If this hypothesis were to prove to be the case, it would have profound implications for how cachexia is defined and how it is treated in cancer patients.

With reference to the definition of cancer cachexia, it has been currently defined as weight loss > 5% or BMI < 20 kg/m^2^ with weight loss > 2% or sarcopenia with weight loss > 2% [53]. However, the present review and the above rationale make a powerful argument for the definition of cancer cachexia to be based on the presence of a systemic inflammatory response, the mGPS, given its consistent thresholds [10]. This can be clarified using a quote by MacDonald in his 2012 review article. ‘The seminal observation by McMillan and colleagues that the presence of dysregulated state as evidenced by a high CRP connotes a dire prognosis has generally been ignored to date and not used to stratify patients in oncology clinical trials. Particularly in the more aggressive tumour types (e.g., pancreas and lung), the future of patients with elevated mGPS is so grim that they should be given precachexia status and offered multimodal therapy which may delay the onset of cachexia and/or death [47]’. More recently, Baracos et al., proposed that the cardinal feature of cachexia was the loss of skeletal muscle [54]. Given that the systemic inflammatory response is a major driver of this loss (supported by the present review), it can readily be argued that the systemic inflammatory response forms the basis of definition of cancer cachexia. Indeed, there is increasing data to support such an approach [55]. Clearly in light of the present review, the systemic inflammation may be combined with a low SMI [25] and/ or combined with performance status [56,57] to better define cachexia.

With reference to the treatment of cancer cachexia, the present review suggests that systemic inflammatory response should be primarily targeted. Unfortunately, to date, few attempts have been made to use systemic inflammation as a therapeutic end-point [58]. More recently, an early phase clinical trial using a multimodal intervention with an anti-inflammatory agent (Ibuprofen, Trondheim, Norway) had a positive effect on the weight and lean body mass and this is now being examined in a phase 3 trial (Trial registration number NCT02330926) in advanced cancer patients [59]. Using a more potent anti-inflammatory, another randomized controlled trial is underway, using bermekimab, which is a humanized antibody to IL-1α [49] and examining its effects on muscles, physical function and appetite in patients with lung, pancreatic or ovarian cancer (MICA trial). If anti-inflammatory treatment given to patients that had evidence of a systemic inflammatory response were proven to prevent further loss of skeletal muscle, this would be a major step forward for the definition and treatment of patients with cancer cachexia. 

A potential management algorithm is shown in Figure 2. On the CT staging of the tumour, there should also be assessment of body composition and laboratory assessment of the systemic inflammatory response. In particular, assessment of SMI and mGPS should be carried out. Such staging of the tumour and host would provide the basis for patient optimization, providing nutritional support and anti-inflammatory agents [60].

This systematic review has some limitations. Firstly, included studies were mainly retrospective and cross sectional. Secondly, the studies were heterogeneous, with various markers of systemic inflammation across a range of various cancers. Thirdly, most of the studies were from single institutions. Large prospective multi-centre follow-up studies involving collaborations among researchers, clinicians, dieticians, physiotherapists, nurses and the pharmaceutical industry are required to generalize the findings of this systematic review and to provide the best patient care. Moreover, how an algorithm could be routinely incorporated into standard radiological imaging software to capture SMI and SMD for clinical reporting remains to be established. At present, it is not clear whether muscle loss from cancer can be differentiated from purposeful weight loss using CT.

## 5. Conclusions

The present systematic review shows low SMI and low SMD to be consistently associated with measures of systemic inflammatory response, including CRP, albumin, mGPS and NLR, in patients with cancer. These observations have implications for the definition and treatment of cancer cachexia which should include measures of the systemic inflammatory response. Once the technical hurdles can be overcome, reporting of SMI should be considered as a routine part of radiology reporting because of its clinical significance.

## Figures and Tables

**Figure 1 cancers-11-01304-f001:**
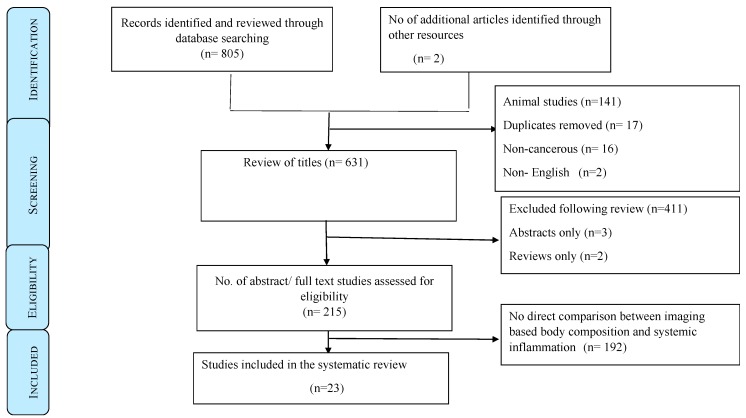
Preferred reporting items for systematic review protocol flow diagram.

**Figure 2 cancers-11-01304-f002:**
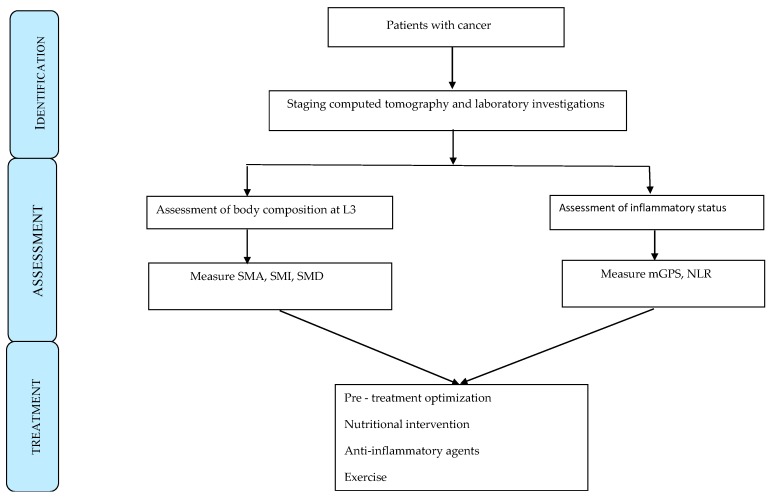
Management algorithm of pre-treatment assessment in patients with cancer. SMA = Skeletal muscle area, SMI = Skeletal muscle index, SMI = Skeletal muscle density, mGPS = modified Glasgow prognostic score, NLR = neutrophil lymphocyte ratio.

**Table 1 cancers-11-01304-t001:** Framework based on modified Glasgow Prognostic score (mGPS).

mGPS	Biochemical Markers		Cachexia Stage
	CRP (mg/L)	Albumin (g/L)	
0	<10	≥35	No cachexia
0	<10	<35	Undernourished
1	>10	≥35	Pre-cachexia
2	>10	<35	Refractory cachexia

CRP = C-reactive protein.

**Table 2 cancers-11-01304-t002:** Characteristics of included studies.

Authors (Year)	Reported STROBE Checklist Points	Type of Study	*n* (F/M)	Country	Cancer Studied	Cancer Stage	Level of Analysis	Systemic Inflammation	Comments
**DEXA**									
Ellegård et al., 2009 [14]	20	Prospective cross-sectional	132 (46/86)	Sweden & New Zealand	Gastrointestinal	Advanced inoperable	Whole body	CRP, Albumin	Low SMI directly associated with elevated CRP and low albumin (*p* < 0.05).
Wallengren et al., 2014 [15]	19	Prospective longitudinal	471 (212/259)	Sweden	Gastrointestinal, pancreatic-biliary	Advanced inoperable	Whole body	CRP, Albumin	Low SMI directly associated with elevated CRP (*p* < 0.001).
Chambard et al., 2018 [16]	20	Prospective cross-sectional	64 (16/48)	France	Non-small cell Lung	Advanced inoperable	Whole body	CRP, Albumin, WCC	Low SMI directly associated with elevated CRP (*p* < 0.05) & WCC (*p* < 0.001).
**CT**									
Richards et al., 2012 [23]	20	Prospective cross-sectional	174 (79/95)	United Kingdom	Colo-rectal	Primary operable	L3	CRP, Albumin, mGPS, NLR	Low SMI (34%) directly associated with elevated mGPS (32%) (*p* < 0.001)
Itoh et al., 2013 [33]	19	Retrospective cross-sectional	190 (44/146)	Japan	Hepatocellular	Primary operable	L3	Albumin	Low visceral fat area associated with sarcopenia (*p* < 0.001) and low albumin (*p* < 0.005)
Reisinger et al., 2016 [30]	17	Prospective cross-sectional	87 (31/56)	Netherlands	Colo-rectal	Primary operable	L3	CRP, mGPS	Low SMI associated with elevated CRP (*p* = 0.05).
Rollins et al., 2016 [17]	18	Retrospective cross-sectional	229 (105/124)	United Kingdom	Pancreatic-biliary	Advanced inoperable	L3	CRP, Albumin, mGPS, NLR	Low SMI and SMD associated with elevated CRP (*p* < 0.05), low albumin (*p* < 0.001) and elevated NLR (*p* < 0.01).
Malietz et al., 2016 [19]	19	Prospective longitudinal	763 (306/457)	United Kingdom	Colo-rectal	Primary operable	L3	Albumin, NLR	Low SMI (65%) and low SMD (84%) associated with NLR > 3 (61% & 57%) (*p* < 0.001) and low albumin (28% each) (*p* = 0.01).
Kim et al., 2016 [31]	20	Retrospective cross-sectional	186 (30/156)	South Korea	Small cell lung	Primary operable	L3	CRP, Albumin, mGPS, NLR	Low SMI associated with elevated CRP (*p* < 0.05), low albumin (*p* < 0.05) and elevated NLR (*p* < 0.01).
Zhuang et al., 2016 [28]	19	Retrospective cross-sectional	937 (207/730)	China	Gastric	Primary operable	L3	Albumin	Low SMI associated with low albumin (*p* < 0.001).
Huang, et al., 2016 [29]	20	Prospective cross-sectional	470 (364/106)	China	Gastric	Primary operable	L3	Albumin	Low SMI associated with low albumin (*p* < 0.001).
Van Di Jik et al., 2017 [18]	19	Prospective cross-sectional	186 (84/102)	Netherlands	Pancreatic	Both operable and inoperable	L3	CRP, Albumin, mGPS	Low SMD associated with low albumin (*p* < 0.01)
Feliciano et al., 2017 (C-SCANS study) [20]	20	Retrospective longitudinal	2470 (1219/1251)	United States, Canada	Colo-rectal	Primary operable	L3	CRP, Albumin, NLR, IL-6	Low SMI associated with elevated CRP (*p* < 0.05), low albumin (*p* < 0.01) and elevated IL-6 (*p* < 0.05)
Srdic et al., 2017 [34]	20	Prospective cross-sectional	100 (33/67)	Croatia	Non-small cell lung	Advanced inoperable	L3	CRP, albumin, mGPS	Low SMI (15% loss of skeletal muscle mass) associated with low albumin (*p* < 0.01)
Kiyotoki et al., 2017 [26]	20	Retrospective cross-sectional	60 All females	Japan	Cervical	Primary operable	L3	CRP, Albumin	Low SMI associated with low albumin (*p* < 0.01).
Serra et al., 2017 [32]	16	Prospective cross-sectional	11 All females	United States	Breast	Primary operable	L4-L5	CRP, Albumin	Significant improvement in muscle strength with resistance training with reduction in inflammatory mediators including CRP.
McSorley et al., 2017 [24]	20	Retrospective cross-sectional	322 (148/174)	United Kingdom	Colo-rectal	Primary operable	L3	CRP, Albumin, mGPS, NLR	Low SMI (47%) and SMD (58%) associated with elevated mGPS (23%) and NLR > 3 (44%) (*p* < 0.01).
Van DiJik et al., 2018 [21]	20	Prospective cross-sectional	97 (30/67)	Canada	Colo-rectal	Primary & metastatic both operable	L3	CRP, Albumin	Low SMI (65%) associated with elevated CRP > 5 mg/dL (74%) (*p* < 0.05).
Okugawa et al., 2018 [35]	20	Prospective cross-sectional	308 (125/183)	Japan	Colo-rectal	Primary operable	L3	CRP, Albumin, NLR, PLR	Low SMI and SMD associated with elevated CRP (*p* < 0.0001) and low albumin (*p* < 0.05).
Dolan et al., 2018 [25]	19	Retrospective cross-sectional	650 (296/354)	United Kingdom	Colo-rectal	Primary operable	L3	CRP, Albumin, mGPS, NLR	Low SMI (44%) and SMD (60%) associated with elevated mGPS (23%) (*p* < 0.001) and NLR > 3 (43%) (*p* < 0.05).
Sueda et al., 2018 [27]	20	Retrospective cross-sectional	211 (77/134)	Japan	Colo-rectal	Primary operable	L3	Albumin, NLR	Low SMI (48%) and SMD (49%) associated with NLR > 3 (41%) with (*p* < 0.05) and *p* < 0.01 respectively.
Basile et al., 2019 [36]	20	Retrospective longitudinal	94 (42/52)	Italy	Pancreatic	Advanced inoperable	L3	CRP, Albumin, NLR	Low SMI & SMD associated with NLR > 5(*p* < 0.001).
Xiao et al., 2019 [22]	20	Retrospective cross-sectional	3262 (1628/1624)	United States	Colo-rectal	Primary Operable	L3	CRP, Albumin, NLR	Low SMI & SMD associated with raised NLR ≥ 5 (*p* < 0.001).

L3 = Lumbar 3 vertebral level, SM I = Skeletal muscle index, SMD = Skeletal muscle density, mGP S = modified Glasgow prognostic score, NLR = Neutrophil lymphocyte ratio

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
