# Peer review of "The Relationship between Imaging-Based Body Composition Analysis and the Systemic Inflammatory Response in Patients with Cancer: A Systematic Review"

_cancers, 2019, doi:10.3390/cancers11091304_

Round 1

Reviewer 1 Report

This is a well written review on a topic of considerable clinical importance. The study was consistent with the STROBE approach to systematic reviews and the discussion was thorough and tailored to clinical practice. There are some comments below that I would think can enhance the review in terms of clarifying definitions and explanation of results. 

Introduction

References missing on a very definitive statement – Line 64 – is it proven/validated that CT and at which point of the torso is a measure that provides reliable evidence of body composition routinely in terms of quality and quantity? Is this methodology approved or investigated in other diseases of wasting? Can you differentiate between cancer-related wasting and anorexia or purposeful weight loss using CT?

Methods

Line 97 and Figure 1 – could the authors please explain in more detail what “not relevant to the topic” covers? Presumably the 411 papers had search terms in the title, but not being relevant does not explain the rationale for exclusion to the reviewer and wider audience. Please amend the text to explain more detail of reasons for exclusion.

Line 109 – could the authors please include references into which studies used particular methodological approaches for ease of identification.

Also, could the authors qualify whether the measurements of cachexia using different and also same methodologies used the same quantitative approach to measure SKI or SMD.

Could the authors please articulate in the methods how SKI and SMD are routinely calculated in these studies? It is not clear if there is a consistent equation/approach used between studies.

Only if there is consistency, should the study cohorts be combined. If different, could these different approaches please be mentioned in the results and further discussed in the discussion.

Results

Abbreviations – many are not common, please refrain from using – SMI or SMD, it would be as easy to write out in full for a ease of reading.

Line 134-137 – a “significant relationship” should be qualified as to the direction of the relationships, saying it was significant does not alert the audience to what was the nature of the relationship. Was there negative relationship? E.g. increased inflammation associated with higher or lower skeletal muscle index. I appreciate it is in the table, but it should be clear if a consistent relationship is found in the written text of the results.

Line 138 – again, what was the nature of the relationship with survival – low skeletal muscle index was associated with shorter survival?

References 25, 29 – could these two papers which measure longitudinal CT imaging, please be expanded in discussion of results. They are mentioned in the discussion, but I missed this point in the results.

Discussion

Do all patients with cancer have CT conducted at the appropriate level? Could the authors discuss how generalisable this type of imaging could be the wider cancer population, not just the gastrointestinal or lung cancers? How difficult would it be to include an algorithm to standard imaging to capture the SKI or SMD for clinicians and therefore translation of these research questions into clinical practice. Could this be included in the discussion.

Line 162 – inoperable cancer is often higher stage, not just increased systemic inflammation

Line 181 – “definition of cancer cachexia to be based on the presence of a systemic inflammatory response” the authors should be more cautious in regards to this statement. Do all patients who present with systemic inflammation also present with cachexia? Perhaps “definition of cancer cachexia could include the presence of systemic inflammation” would be more appropriate.

 Line 187 – Please add in the generic name of the anti-inflammatory agent that had a positive effect on weight/lean body mass and also details of which stage of cancer cachexia to demonstrate relevance to this discussion. Also include the clinical trial number (clinical trial.gov) for future follow-up for readers.

Line 190 – Again, clinical trial number would be useful for the bermekimab trial, which should also be in lower case letters. More details are needed in this sentence about the trial – ie cancer type, definition of cachexia, placebo?

Figure 2 – Could details be included of what measure of tumour staging should be completed – CT at L3 and which systemic inflammation markers – CRP, Albumin, FBC for NLR or PLR. Rather than superscripts a second line could be added to include this information on an informative flow diagram for clinicians.

Minor issue – additional spaces throughout the manuscript, behind commas and in the middle of sentences, please carefully review.

Reviewer 2 Report

The topic is very interesting. I am not convinced by the literature search strategy: for exemple the paper  by Hervochon et al (Ann Thorac Surg 2017; 103:287-295) which fullfils all the scopes of the research was not found. Probably several other works have been published, should be acknowledged and taken into account in this systematic review. 

Round 2

Reviewer 2 Report

I am not convinced of authors' reply to my questions.

Measurement of Psoas muscle at third lumbar vertebra is considered adequate in a large amount of available literature on the topic. 

The references have been revised, but in the authors' opinion studies
employing measurement of psoas muscles areas rather than other measures
of muscle area cannot be included. In their reply letter, authors state
that "The study by Hervochon et al (Hervochon, Bobbio et al. 2017) was
identified in our search strategy. However, it is now recognised that
the measurement of psoas muscles at lumbar 3 level is less reliable and
inferior to measuring all muscles at this level (Baracos 2017) and
therefore, this study was excluded from the review". I am not convinced
that available level of evidence is sufficient to claim inferiority of a
technique over another (see Icard P, Iannelli A, Lincet H, Alifano M.
Sarcopenia in resected non-small cell lung cancer: let’s move to
patient-directed strategies. J Thorac Dis. PMID: 30370098); anyway this
supposed less reliability (in their opinion) does not warrant exclusion
from the review analysis in this kind of study.

Author Response

Dear Reviewer,

Thank you very much for your kind suggestion. Please find our response as below.

REVIEWER 2

Comment

The references have been revised, but in the authors' opinion studies employing measurement of psoas muscles areas rather than other measures of muscle area cannot be included. In their reply letter, authors state that "The study by Hervochon et al (Hervochon, Bobbio et al. 2017) was identified in our search strategy. However, it is now recognised that the measurement of psoas muscles at lumbar 3 level is less reliable and inferior to measuring all muscles at this level (Baracos 2017) and therefore, this study was excluded from the review". I am not convinced that available level of evidence is sufficient to claim inferiority of a technique over another (see Icard P, Iannelli A, Lincet H, Alifano M.Sarcopenia in resected non-small cell lung cancer: let’s move to patient-directed strategies. J Thorac Dis. PMID: 30370098); anyway, this supposed less reliability (in their opinion) does not warrant exclusion from the review analysis in this kind of study.

Point taken. “There is some debate as to whether, the measurement of psoas muscle at lumbar 3 level is less reliable and inferior to measuring all muscles at this level (Baracos 2017, Icard, Iannelli et al. 2018) and therefore, these studies (Hervochon, Bobbio et al. 2017, Okugawa, Toiyama et al. 2019) are considered separately . Using psoas muscle measurement, Hervochon and co-workers, in a cohort of 161 patients with operable NSCLC, reported  that low SMI (total psoas area ≤ 33rd percentile) was significantly associated with elevated CRP (Hervochon, Bobbio et al. 2017). Furthermore, Okugawa and co-workers, in a cohort of 308 patients with operable CRC reported that low SMI (using sex- specific median values of psoas muscle index, male: 286.8 mm2/m2, female: 210.6 mm2/m2) was significantly associated with elevated CRP and low albumin (Okugawa, Toiyama et al. 2019). Therefore, it would appear that skeletal muscle, however assessed from CT scans, is consistently associated with measures of the systemic inflammatory response.” Text amended in the discussion section.

References:

Baracos, V. E. (2017). "Psoas as a sentinel muscle for sarcopenia: a flawed premise." J Cachexia Sarcopenia Muscle 8(4): 527-528.

Hervochon, R., A. Bobbio, C. Guinet, A. Mansuet-Lupo, A. Rabbat, J. F. Regnard, N. Roche, D. Damotte, A. Iannelli and M. Alifano (2017). "Body Mass Index and Total Psoas Area Affect Outcomes in Patients Undergoing Pneumonectomy for Cancer." Ann Thorac Surg 103(1): 287-295.

Icard, P., A. Iannelli, H. Lincet and M. Alifano (2018). "Sarcopenia in resected non-small cell lung cancer: let’s move to patient-directed strategies." Journal of Thoracic Disease: S3138-S3142.

Okugawa, Y., Y. Toiyama, A. Yamamoto, T. Shigemori, A. Kitamura, T. Ichikawa, S. Ide, T. Kitajima, H. Fujikawa, H. Yasuda, Y. Okita, J. Hiro, T. Araki, D. C. McMillan, C. Miki and M. Kusunoki (2019). "Close Relationship Between Immunological/Inflammatory Markers and Myopenia and Myosteatosis in Patients With Colorectal Cancer: A Propensity Score Matching Analysis." JPEN J Parenter Enteral Nutr 43(4): 508-515.

We look forward to your reply in due course.

Round 3

Reviewer 2 Report

I suggest acceptance without further corrections.

Author Response

Many thanks for comments. We appreciate your comment of " I suggest acceptance without further corrections."